# Numerical and Field Investigations of Acoustic Emission Laws of Coal Fracture under Hydro-Mechanical Coupling Loading

**DOI:** 10.3390/ma15196510

**Published:** 2022-09-20

**Authors:** Jie-Fang Song, Cai-Ping Lu, Zhao-Wei Zhan, Hai-Feng Cui, Yan-Min Wang, Jian-Hua Wang

**Affiliations:** 1Key Laboratory of Deep Coal Resource Mining, Ministry of Education, School of Mines, China University of Mining and Technology, Xuzhou 221116, China; 2Xiaoyun Coal Mine, Jining Energy Group, Jining 272000, China

**Keywords:** hydro-mechanical coupling, failure mechanism, confining pressure, acoustic emission, moment tensor inversion

## Abstract

**Abstract:**

Taking coal under hydro-mechanical coupling as the research object, the discrete element software PFC3D (particle flow code) was used to analyze the relationships among the force, acoustic emission (AE), and energy during coal fracture. Based on the moment tensor (MT) inversion, we revealed the AE event distribution and source type during crack initiation and propagation until the final failure of coal. Meanwhile, we examined the relationships among the stress, number and type of cracks, magnitude, K_E_, and b value of AE under different water and confining pressures. The results show that the numerical simulation can effectively determine the microscopic damage mechanism of coal under different conditions. Moreover, the rupture type of the numerical simulation is consistent with the field investigations, which verifies the rationality of the simulation. These research results can provide reference for safety production evaluation of water inrush mines.

**Highlights:**

## 1. Introduction

With the continuous increase in the mining depth of coal mines, geological conditions have become particularly complex [1]. Discontinuous joints such as microcracks in a rock stratum provide necessary locations for the storage and migration of groundwater [2,3]. In the seepage process, the generated water stress affects the stress field of a coal and rock mass [4]. Under the superposition of multiple stress fields, the frequency and intensity of catastrophic accidents such as rock bursts and water inrush remarkably increase [5,6]. When a water inrush accident occurs, the overall strength and plastic deformation of a coal pillar significantly change, which has a notable impact on the support capacity and resistance of the roadway confining pressure system, thereby affecting the width of the coal pillar [7].

Studies on the mechanical properties of coal and rock under hydro-mechanical coupling have mainly focused on the interaction between the seepage and stress fields, and the research methods have primarily included laboratory experiments and numerical simulations. For example, Hui et al. [8] conducted a series of laboratory acoustic emission (AE) tests to study the influence of fluid-filled rock joints on the propagation and attenuation of stress waves. Gardner et al. [9] investigated the interaction relationship between a rock mass and a fluid based on the three-dimensional discrete element and lattice Boltzmann methods. For the purpose of improving the calculation efficiency and accuracy, an extensive discussion on aspects that still need to be optimized was conducted. Chen et al. [10] studied the characteristics of the permeability evolution during sandstone failure under triaxial compression by experiments and simulations. Cai et al. [11] analyzed the evolution of the crack growth, coordination number, volumetric strain, and permeability under hydro-mechanical coupling in sandstone and established a synergistic mechanism among the microstructure, accumulated damage, and permeability evolution. Liu et al. [12] analyzed various parametric changes of a rock mass with hydro-mechanical coupling under unloading conditions. Wang et al. [13] investigated the mechanical characteristics and permeability evolution of red sandstone under hydro-mechanical coupling. Yao et al. [14] analyzed the mechanical properties and failure precursor characteristics of a coal and rock system caused by water content using AE and infrared radiation. Wang et al. [15] studied the deformation characteristics of granite under hydro-mechanical coupling and analyzed the changes in the permeability before and after expansion under different confining pressures. Zhang et al. [16] studied the influence of water and confining pressures on the permeability of fractured sandstone. Many scholars have studied the changes of rock mechanical parameters under the hydro-mechanical coupling loading in the laboratory, but there are few studies on the influence of seepage under different water and confining pressure [17,18]. In addition, few scholars have focused on the source rupture type in the process of rock seepage through AE.

In recent years, AE has been commonly used in the analysis of coal and rock failure. As an accompanying phenomenon in this process, it contains substantial information about the internal damage process of rocks, which helps to understand the mechanism of rock failure [19,20,21]. For instance, Shimizu et al. [22] discussed the effects of different fluid viscosity and particle size distributions on fracture propagation. Jia et al. [23] analyzed the relationship between AE and spatial fractal dimensions under different water soaking times. Ye et al. [24] obtained the relationships among AE events and the joint angle, water injection pressure, stress, and deformation amount in the process of a rock slip instability. Makhnenko et al. [25] observed that AE activity coincides with the onset of inelastic response in a fluid-saturated rock and explored the relationship between the clustering of AE events and inhomogeneous deformation. Li et al. [26] applied AE to dynamic disaster monitoring of coal and rock systems. Li et al. [27] concluded that there is a good positive correlation between the change in coal AE events and the water pressure curve under the action of triaxial hydraulic coupling. Chitrala et al. [28] reported the changes in the AE during hydraulic pressure damage of sandstone under different applied stresses. Chen et al. [29] used AE to analyze the evolution characteristics of granite damage during uniaxial compression and established a relationship between the permeability and the confining pressure. Makoto et al. [30] studied the relationship between the AE and fluid pressure during the failure of the Eagle Ford Shale, and analyzed the main types of AE. Lu et al. [31] successfully reproduced the stress redistribution, AE event evolution, permeability change, and formation of water inrush channels during a mining process. Jiang et al. [32] analyzed the AE event, energy, peak frequency, and crack type under the action of hydraulic fracturing in a layered rock. Most scholars widely use AE to locate the crack position of rock, and then study the force of the specimen [33]. However, there are few studies on the focal mechanism of AE, and analysis of the rupture form of rock/coal mass under different stress conditions is still lacking.

In summary, most studies mainly focused on the basic mechanical parameters of coal and rock masses under the action of hydro-mechanical coupling. However, they did not conduct detailed investigations on the focal mechanism of this action, and rarely used an energy method to evaluate the impact tendency of coal under hydro-mechanism coupling [34,35]. To better understand the characteristics of AE and its impact tendency under hydro-mechanism coupling, this study used the PFC3D software to analyze the relationships among the force, AE distribution (type), energy, and b value in the failure process of a coal sample under different water pressures. In addition, it is consistent with the results of our field investigation. These results are highly significant for guiding the safe production of coal mines.

## 2. Damage Mechanism of Coal under Hydro-Mechanism Coupling

To ensure no shear failure occurs during the loading of a rock, the limit value of the rock force can be obtained by the Mohr–Coulomb criterion [36] as follows:(1)$$\left|\tau \right|=C+\gamma_{s}\sigma$$
where $\left|\tau \right|$ is the shear stress of the rock, MPa, C is the internal cohesion, Pa, and $\gamma_{s}$ is the internal friction coefficient, which can be also expressed as
(2)$$\gamma_{s}=\tan\varphi$$

It is assumed that the minimum principal stress on the rock is $\sigma_{3}$. When the rock is in the critical limit state, the maximum principal stress, $\sigma_{1}$, can be calculated. Owing to the existence of water pressure in the rock pores, a part of the confining pressure and vertical stress is offset; therefore, the water pressure eliminates the confining pressure (reducing the servo stress). The triaxial stress state of the rock under water pressure is shown in Figure 1.

When the Mohr–Coulomb circle is tangent to the envelope curve, the length of the right-angle side of the grey triangle area is $\left(\lambda_{1}\sigma_{1}-\lambda_{2}\sigma_{3}\right)/2$, and the hypotenuse length is $C/\tan\varphi +\left(\lambda_{1}\sigma_{1}+\lambda_{2}\sigma_{3}\right)/2$. Based on trigonometric functions,
(3)$$\left(\lambda_{1}\sigma_{1}-\lambda_{2}\sigma_{3}\right)/2=\left[C/\tan\varphi +\left(\lambda_{1}\sigma_{1}+\lambda_{2}\sigma_{3}\right)/2\right]\sin\varphi$$

Combined with Equation (2), the maximum principal stress under the critical stress state of the sample can be expressed as
(4)$$\sigma_{1}=\frac{2C+\lambda_{2}\left(\gamma_{s}+\sqrt{\gamma_{s}^{2}+1}\right)\sigma_{3}}{\lambda_{1}\left(\sqrt{\gamma_{s}^{2}+1}-\gamma_{s}\right)}$$

In the experiments, when the internal cohesion and the friction angle have certain values, because $\lambda_{1}$ and $\lambda_{2}$ are less than 1, the stress–strain curve of the sample under water pressure is above that of the sample without water pressure; however, the peak stress in the former is smaller. This is attributed to the effect of the pore water pressure reducing the effective average stress of the rock, making it easier for the rock to reach the ultimate strength when the deviatoric stress remains constant.

## 3. Modeling Methodology

### 3.1. Engineering Background

A water inrush accident occurred in the Xiaoyun coal mine in Shandong province on 11 September 2018. The coal body of the 1318 working face was severely affected by the water pressure of proximately 3 MPa, and the soaking time was 100 days. A sample was taken from the soaking water coal body of the above working face, which has a simple structure and is relatively stable. To make the sample moisture non-volatile, it was stored in plastic wrap before experiments and processed into a cylinder with a diameter of 50 mm and a length of 100 mm. An MTS Landmark 370.50 rock testing machine was used for loading.

### 3.2. Model Setup

In the discrete element model, the core of the fluid flow algorithm is the topological structure formed by pipe domains. As shown in Figure 2, many domains that can store the pore water pressure are formed between the particles, and these domains are represented by red squares. A fracture between adjacent particles is regarded as a seepage pipe to simulate the flow of water. A pipeline is represented by a yellow line. When there is a pressure difference between adjacent domains, the fluid flows through the pipe to them.

To study the mechanical characteristics and instability mechanism of coal under the action of seepage, a model was set up, which is shown in Figure 2. For the model, the standard size (Φ50 mm × 100 mm) was adopted, and a total of 3117 particles were generated. The radius was 2–2.5 mm. The linear contact bond model was used between particles, and the servo mechanism was used to apply different confining pressures around the model. When the confining pressure reached the target stress, the particles in contact with the wall were fixed. A high-pressure water area was applied within the range of 0–10 mm in the *z*-axis direction at the bottom of the model. Under the action of the pressure difference, a water body seeped from the bottom to the top of the model. At this time, the water pressure is close to linear distribution along the longitudinal direction, which can be regarded as the equilibrium state of hydro-mechanical coupling. After the balance was reached, the upper and lower walls were controlled to apply an axial pressure at a speed of 0.03 m/s, and the loading was stopped when the residual stress reached 70% peak stress.

The model parameter calibration was based on the uniaxial compressive strength, strain, and failure form of the specimen. Based on Figure 3, the peak stresses from the experiments and numerical simulations are 6.09 MPa and 6.05 MPa, respectively, and the corresponding strains are 1.93% and 1.95%. The difference between them is relatively small, and the failure modes are also similar. From the variation trends of the AE events, the microscopic properties inside the coal body are changed owing to the softening effect of water. When the strain is approximately 5%, the number of AE events shows remarkable fluctuations, which is followed by a quiet period. When the strain reaches 1.6%, the AE events abruptly increase, subsequently cracks gradually penetrate to form macroscopic cracks, and, finally, the number of AE reaches the maximum. Based on the above analysis, the rationality of the microparameter values can be verified, which are listed in Table 1. 

### 3.3. MT Calculation in the PFC

In seismology, MT is a mathematical representation of fault motion and an important tool for source characterization. We use *M* to represent the source MT, which is a 3 × 3 symmetric matrix with nine couples and six independent components, as shown in Figure 4. 

Figure 4 shows the force couples in the Cartesian coordinate system, which is related to source strength and fault direction. We can further decompose the MT into elementary sources, e.g., compression, shear, and tensile. Then, we can intuitively obtain the rupture type of A.E. Hudson’s source type plot [37,38], which is a way to vividly represent decomposition of an MT into isotropic, compensated linear vector dipole (CLVD) and double-couple (DC) components. The MT decomposition is shown in the following Equations (5) and (6):(5)$$\begin{array}{c} M=M_{DC}+M_{iso}+M_{v-clvd}\\ =M_{1}\left(\phi_{S}\right)+M_{2}\left(\phi_{D}\right)+M_{iso}+M_{v-clvd} \end{array}$$
(6)$$\begin{array}{c} \left(\begin{array}{ccc} M_{11} & M_{12} & M_{13}\\ M_{21} & M_{22} & M_{23}\\ M_{31} & M_{32} & M_{33} \end{array}\right)=\left(\begin{array}{ccc} -\frac{1}{2}\left(M_{22}-M_{11}\right) & M_{12} & 0\\ M_{21} & \frac{1}{2}\left(M_{22}-M_{11}\right) & 0\\ 0 & 0 & 0 \end{array}\right)+\left(\begin{array}{ccc} 0 & 0 & M_{13}\\ 0 & 0 & M_{23}\\ M_{31} & M_{32} & 0 \end{array}\right)\\ +\frac{1}{3}\left(M_{11}+M_{22}+M_{33}\right)\left(\begin{array}{ccc} 1 & 0 & 0\\ 0 & 1 & 0\\ 0 & 0 & 1 \end{array}\right)+\frac{1}{3}\left(\frac{1}{2}\left(M_{11}+M_{22}\right)-M_{33}\right)\left(\begin{array}{ccc} 1 & 0 & 0\\ 0 & 1 & 0\\ 0 & 0 & -2 \end{array}\right) \end{array}$$

The PFC 5.0 software (Itasca Consulting Group, Inc., Minneapolis, MN, USA) can directly obtain the force magnitude, direction, and displacement of particles, and then obtain the MT according to the definition of AE. The calculation process of the AE in PFC is shown in Figure 5.

An AE event is composed of particles around microcracks in PFC 5.0 software. When the contact between two particles is broken, a crack will occur. Two particles in contact with the crack and particles in contact with these two particles are marked as an AE event [39,40]. From Figure 5, (a) is the AE start stage, (b) is the AE end stage, and the range delineated by the red circle is the AE event. Through MT decomposition, tensile, shear, and compression failure mode can be obtained. The AE rupture mode corresponding to PFC 5.0 numerical simulation software is shown in Figure 6.

Similar to earthquakes, the intensity of AE can also be expressed by magnitude. In PFC software, the AE intensity is calculated according to the strain energy (*E_k_*) change before and after the event [41]. For each step of calculation, the *E_k_* of all particles in the AE event can be written as Equation (7):(7)$$E_{k}=\sum_{i=1}^{n}\frac{1}{2}\left(\frac{{\left(F_{n}^{l}\right)}^{2}}{k_{n}}+\frac{{\left\|F_{s}^{l}\right\|}^{2}}{k_{s}}\right)$$
where $F_{n}^{l}$, $F_{n}^{s}$, $k_{n}$, and $k_{s}$ are normal stress, shear stress, normal stiffness, and shear stiffness, respectively; *n* is the number of contacts within the AE range. For an AE event, its magnitude can be measured by the change of *E_k_* at the beginning and end of AE [42]:(8)$$M_{e}=\frac{2}{3}\left(\log\Delta E_{k}^{}-6.0\right)$$

Based on the knowledge of seismology, the relationship between the magnitude and the frequency is
(9)$$\log N=a-bM_{e}$$
where *M_e_* is the magnitude, *N* is the frequency, and a and b are constants. The *b* value can be used to evaluate the damage degree of a rock mass and the size of a crack. A large *b* value implies a large proportion of small-sized cracks in the rock mass and a slow initiation and development of new cracks.

## 4. Mechanical Characteristics and Failure Mechanism of Water-Soaked Coal

### 4.1. AE and Stress Changes during Failure

In the simulations, the AE and the energy were monitored to investigate the characteristics of coal failure. To study the coal failure mechanism under hydro-mechanical coupling, the simulations described in Section 4.1 were taken as an example, and the confining and water pressures were set as 1 MPa and 2.5 MPa, respectively. The microscopic characteristics of the AE distribution (type) and energy change during the failure process were analyzed in detail.

#### 4.1.1. Relationship between AE and Stress

To study the coal failure characteristics, the fish language and the history module were used to monitor the vertical stress, strain, AE, energy, etc. Simultaneously, the monitoring circle module was used to record the parameter, and the results are shown in Figure 7 and Figure 8.

The simulated stress–strain curve, cracks, and locations of the AE sources during uniaxial compression of water-soaked coal are shown in Figure 7. The coal failure process can be divided into four stages. The first stage is the compaction stage, in which the stress gradually increases without cracks. The second stage is the crack initiation stage, and the crack initiation stress is 5.33 MPa. The crack distribution is mainly observed in the lower part of the model, and the increase in the number of shear cracks is significantly greater than that in the number of tensile cracks. The third stage shows a notable increase in the cracks. In this stage, the increase rate of the shear cracks is much larger than that of the tensile cracks, and microcracks gradually develop upward. However, the damage degree of the lower part of the entire sample is much greater than that of the upper part, and the peak stress reaches 7.8 MPa. The fourth stage is the post-peak stage, in which the number of cracks sharply increases, the cracks gradually penetrate, and, finally, the entire instability occurs.

By depicting the distribution of the micro contact force in the particle system, the macro mechanical mechanism can be revealed. Figure 8 shows the contact force between particles in the model, which can better reflect the force in different directions of the model. From Figure 8, as the axial stress increases, the internal contact force shows a gradually increasing trend. The shape of the contact force gradually evolves from “spherical” to “elliptical”, the stress levels at different positions gradually change, and at the final stage of failure, the local stress abruptly increases. The vertical stresses at monitoring points F1–F4 are consistent with the stress–strain curve. Except for at point F1, closeness to the water injection area implies a high vertical stress. Based on the change in the yy-stress, when the strain is 1.8%, the horizontal stress remarkably changes, and the model shows an instability at this stage.

#### 4.1.2. AE Event Distribution and Field Verification

In the loading process, a total of 509 AE events were recorded. Figure 9 shows the T–K diagram of the MT obtained using the standard inversion method.

From Figure 9, at the four stages of loading, total of 15, 77, 65, and 417 AE events are recorded, respectively. The distribution of the AE events is relatively scattered, and most of them are distributed in the shear area. At stages (a) and (b), 15 and 77 events are generated, respectively, with magnitudes of between −6.68 and −5.3. The corresponding stresses at the end of these two stages are 5.32 MPa and 7.4 MPa, which are approximately 68% and 94% of peak stress, respectively. The AE events during this period are mainly dominated by the water pressure. At stages (c) and (d), the axial stress gradually reaches the peak strength of the coal. During this period, the effect of the axial stress is far greater than that of the water pressure, and the number of AE events sharply increases (mainly shear failure), which leads to instability of the sample.

#### 4.1.3. Proportions of Source Type 

To study the types of AE events in coal under the action of hydro-mechanical coupling, the number and proportion of tensile, shear, and implosion source types at different failure stages were quantitatively counted. The results are shown in Figure 10.

From Figure 10, as the axial stress increases, the numbers of implosions and shears sharply increase; however, their proportions first increase and subsequently decrease. The entire damage is dominated by shear cracks. In stages (a) and (b), the proportions of shear and tension source types become larger, from 69% and 25% to 70% and 29%, respectively. Tensile failure increases in a slightly larger proportion than shear cracks. Observing the AE distribution shows that the damage is mainly concentrated in the lower part of the model. The particles in the water pressure area are subjected to tension. Simultaneously, tensile failure is prone to occur under axial stress. With the increase in the axial stress, the influence of the water pressure gradually weakens. Therefore, in stages (c) and (d), the proportion of tensile source type slightly decreases, whereas that of implosion source type increases to 8%. In summary, the water pressure mainly affects the early stage of loading and is primarily influenced by the axial stress in the later failure process.

#### 4.1.4. Energy Change 

Coal failure process is accompanied by generation, accumulation, dissipation, and release of energy, and the evolution of energy is closely consistent with the expansion of internal cracks in coal. Using the energy module of the PFC3D, the energy variation was recorded during coal failure process.

From Figure 11, the kinetic energy tends to increase in the initial stage of loading. The main cause is the force unbalance between some particles under the action of the axial pressure, which destroys the linear parallel bonds between them. There is no increase in the slip energy within a certain period. When the vertical stress of the coal body reaches approximately 50% peak stress, the kinetic and slip energies gradually increase. When the strain reaches 2%, the model slips overall. At this time, the slip and kinetic energies gradually reach maximum. The total strain energy is consistent with the overall change trend of the stress–strain curve.

### 4.2. Effects of Water and Confining Pressures on Coal

The confining and water pressures were set as 0.5, 1.5, and 2.5 MPa, respectively. The specific settings are summarized in Table 2.

#### 4.2.1. Failure Characteristics

A total of 12 models were established, and each model number and its corresponding parameters are listed in Table 2. To analyze the internal damage of coal under the effects of different water and confining pressures, the AE events distribution of the different models were examined, and the results are shown in Figure 12.

From Figure 12, it is seen that the confining and water pressures significantly affect the failure pattern of coal. When the model is damaged, the AE events are mainly concentrated in the water pressure area. Based on the comparative analysis of P-1–P-4, P-5–P-8, and P-9–P-12, as the water pressure increases with the same confining pressure, the internal damage and high-energy events increase, and most of the AE events are mainly concentrated in the lower part of the model. When the water pressure is kept constant, the model accumulates more elastic energy before it reaches failure with increasing confining pressure. When the model is damaged, more high-energy events are generated.

From Figure 13, the force of the model has a large anisotropy under the effects of the confining, water, and axial pressures. As the water pressure increases, the number of contacts in the lower part is much smaller than that in the upper part, mainly owing to the greater damage in the lower part. As the confining pressure increases, the number of contacts increases significantly. The maximum number of contacts is increased from 28 to 29, and the influence of the water pressure is gradually weakened, indicating that the strengthening of a support can help protect the stability of the roadway.

#### 4.2.2. Relationship between AE and Stress–Strain Curve

To analyze the effects of different water and confining pressures on the overall strength of the model, the stress–strain curves of the P-1–P-12 models were examined, and are shown in Figure 14.

Based on Figure 14, when the water pressure remains constant, a large confining pressure implies a high peak strength of the sample and a large deformation before failure. When the confining pressure is constant, a large water pressure implies a low peak strength and a small axial deformation before the peak; however, the slope increases, which is consistent with the analysis results of Equation (3).

Figure 15 shows the changes in the numbers of shear and tensile cracks during the failure of different models. The number of shear cracks is approximately twice that of tensile cracks. As the water pressure increases before uniaxial compression, more cracks are generated during the seepage process, and the number of shear cracks is greater than that of tensile cracks. In the uniaxial compression process, when the confining pressure is small, the water pressure has a significant effect on the number of AE events (N_P-2_ ≈ 1.23N_P-3_ ≈ 2.01N_P-4_). When the confining pressure is high, the effect of the water pressure significantly weakens (N_P-10_ ≈ 1.05N_P-11_ ≈ 1.4N_P-12_), and the degree of weakening significantly strengthens.

#### 4.2.3. Frequency–Magnitude Curve and *b* Value

Smaller earthquakes occur more frequently than large earthquakes [43]. This trend can be expressed by the magnitude–frequency relationship and evaluated by *b* value.

Figure 16 shows that there is a good linear relationship between the water and confining pressures and the *b* value. If the confining and water pressures are high, the value of *b* is small, which proves that the large-scale cracks inside the model gradually increase and rapidly and unsteadily expand, and the AE distribution becomes more diffusing. Therefore, as the water and confining pressures increase, the internal stress concentration of the model rises. When cracks occur inside the model, microcracks are more likely to grow and penetrate large-scale cracks. Simultaneously, a high confining pressure implies a large number of AE events inside the rock mass. Thus, the water and confining pressures have remarkable impacts on the AE of the coal body.

### 4.3. Burst Tendency of Coal under Hydro-Mechanism Coupling

There is a certain relationship between the dissipated and strain energies during the uniaxial compression of coal, which is shown in Figure 17.

A schematic of the energy burst tendency is shown in Figure 17, where F1 represents the difference between the elastic energy accumulated before the peak and the dissipated energy consumed during the plastic deformation and crack development of the coal. F2 represents the energy consumed when the coal is damaged after the peak, i.e., F1/F2 is an energy burst tendency indicator of the coal. In the energy module part of the PFC3D 5.0 software (Itasca Consulting Group, Inc., Minneapolis, MN, USA), the total strain energy includes the strain energy ($E_{c}$) and the bond strain energy ($E_{pb}$), and the total dissipated energy includes the frictional energy ($E_{\mu }$) and the kinetic energy ($E_{k}$). Among them, the slip energy remains equal to zero until the bond breaks. It can be seen that the particle flow can monitor the change in the energy in the deformation and failure process of the model in real time. Therefore, in the full stress–strain curve, the elastic and dissipation energies at any time can be expressed in PFC3D as
(10)$$\left\{\begin{array}{l} U_{i}^{e}=E_{pb}+E_{c}\\ U_{i}^{d}=E_{f}+E_{k} \end{array}\right.$$

After deducting the energy consumed by the plastic deformation of the rock and crack development, the burst tendency of the coal body can be determined using the energy method as follows:(11)$$\left\{\begin{array}{l} F_{1}=U_{peak}^{e}+U_{peak}^{d}\\ F_{2}=U_{peak}^{e}+U_{residual}^{e}\\ K_{E}=F_{1}/F_{2} \end{array}\right.$$

Figure 18 shows the calculation results of the burst tendency of coal under different confining and water pressures. When the water and confining pressures are 0 MPa and 1.5 MPa, respectively, the coal body has the lowest burst tendency, and *K_E_* is 2.4. When the water and confining pressures are 1.5 MPa and 0 MPa, respectively, the coal body has the highest burst tendency, and *K_E_* is 2.67. From the *K_E_* distribution, when the water pressure remains constant, as the confining pressure increases, *K_E_* becomes smaller. When the confining pressure remains constant, as the water pressure increases, *K_E_* becomes larger. Therefore, the burst tendency under different conditions can be accurately determined.

## 5. Onsite Investigations

The 1314 working face is located at the level of −790 m; the mining seam is the #3 coal with the thickness of 1.5–3.7 m, and the buried depth is 648–700 m. The roadway is arranged along the roof of the #3 coal seam. The coal seam structure of the working face is simple, with an inclination angle of 13°–26°.

On 11 September 2018, a water inrush accident occurred in this working face, and the coal was immersed in a water pressure of approximately 3 MPa for approximately 100 days, which had a certain impact on the stability of the roadway under the influence of water. By counting MS events during soaking, the Hudson’s source type is obtained according to focal mechanism inversion, as shown in Figure 19. There are a wide range of failure types in the coal, mainly shear and tensile cracks, and its failure mainly occurs in the type of shear cracks. The field observation results are consistent with the simulation conclusions.

In terms of forces, a shear crack can be expressed by two perpendicular force dipoles with zero angular momentum. Therefore, the shear failure source is usually called DC. In order to intuitively show the internal damage form of the coal, several MS events around the working face were selected for detailed analysis. Three types of beachballs, full, deviatoric, and DC, were drawn, as shown in Figure 20. It is found that the shear component accounts for a large proportion in the above MS events, and it is concluded that the force of the shear component is the main factor leading to the failure of the coal. In summary, the failure of coal under water pressure is mainly dominated by shear source, mixed with a small amount of other components, which is strictly consistent with the simulation conclusions. From Figure 21, several fault plane solutions were solved, including two normal fault, three reverse fault, and one normal oblique slip, most of which face the interior of the working face.

## 6. Conclusions

(1)Affected by hydro-mechanical coupling, the damage degree of coal in the flooded part is relatively large, and its damage mainly occurs as shear cracks; this is verified by field observation. The closeness of coal to the water injection area implies a high vertical stress.(2)Monitoring of the energy changes shows that the kinetic and slip energies increase slightly in the initial stage of loading. When the peak stress is greater than 50%, a sharp energy increase occurs, and the increase rate becomes increasingly high.(3)There is a good linear relationship between the water pressure, confining pressure, and *b* value. With increasing water and confining pressures, the damage degree and AE energy inside the model increase, the value of *b* reduces, and most of the AE events are mainly concentrated in the water injection area. When the confining pressure is low, the water pressure has a significant effect on the number of AE events; otherwise, the effect of water pressure gradually weakens.(4)When the water pressure is constant, as the confining pressure increases, *K_E_* decreases. When the confining pressure is kept constant, as the water pressure increases, *K_E_* increases.

## Figures and Tables

**Figure 1 materials-15-06510-f001:**
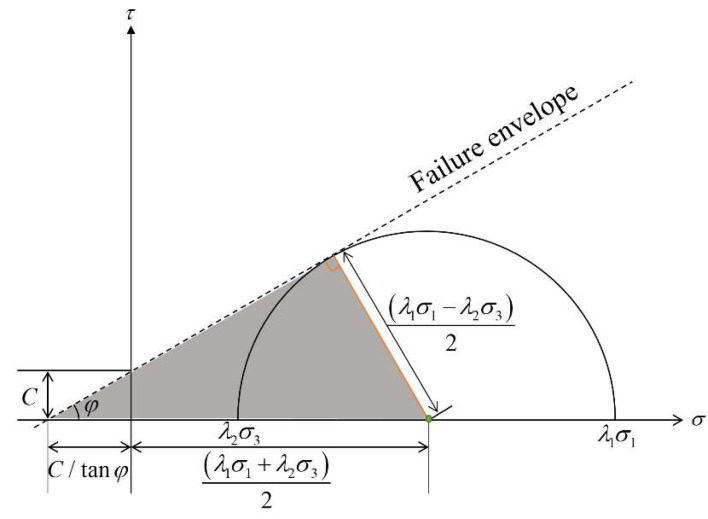
Schematic of critical stress state under influence of water pressure based on Mohr–Coulomb criterion.

**Figure 2 materials-15-06510-f002:**
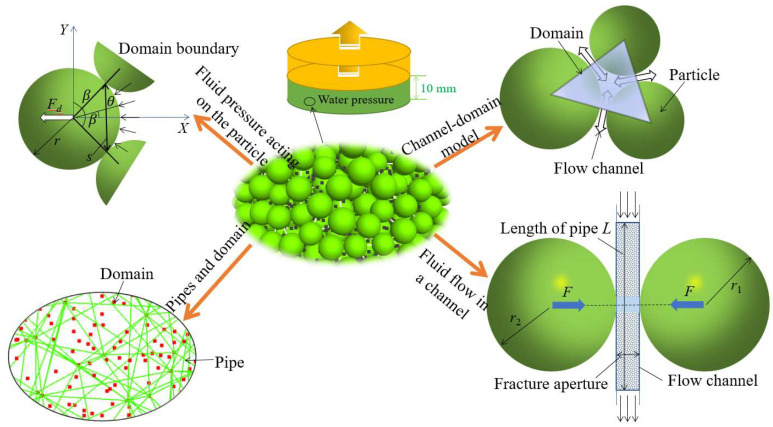
Schematic of incorporation of fluid coupling.

**Figure 3 materials-15-06510-f003:**
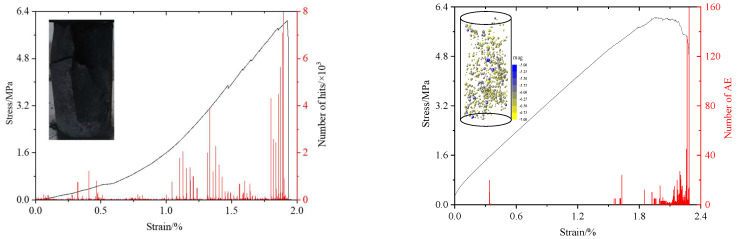
Variation curves of vertical stress and AE impact number of coal sample during uniaxial compression. The red and black lines represent the AE number and the stress-strain curve, respectively. The upper left corner figure is the failure figure of laboratory and simulated respectively.

**Figure 4 materials-15-06510-f004:**
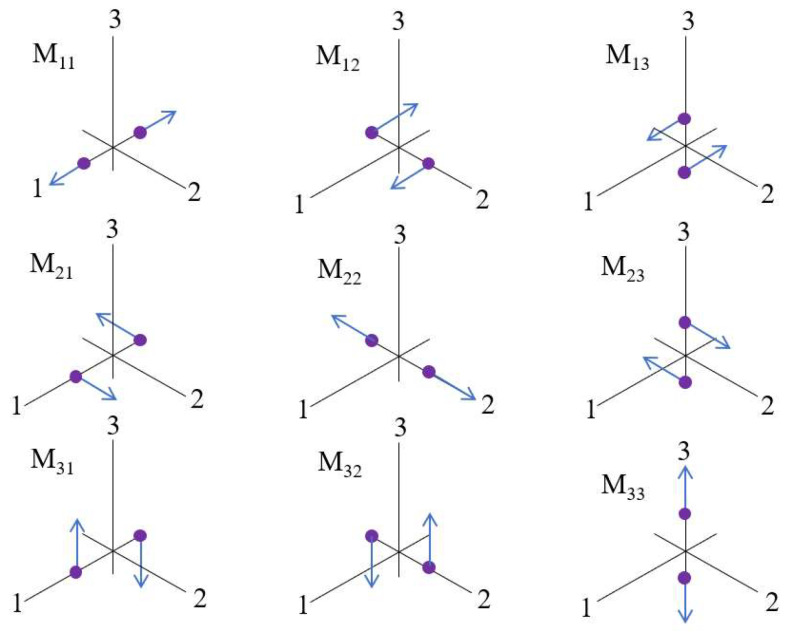
The nine force couples representing the components of MT.

**Figure 5 materials-15-06510-f005:**
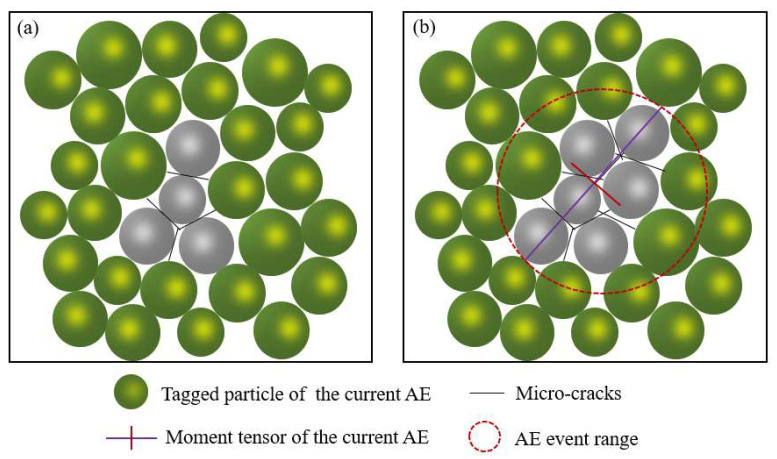
Generation process of an AE event: (**a**) is the start of an event and (**b**) is the finish state of this event.

**Figure 6 materials-15-06510-f006:**
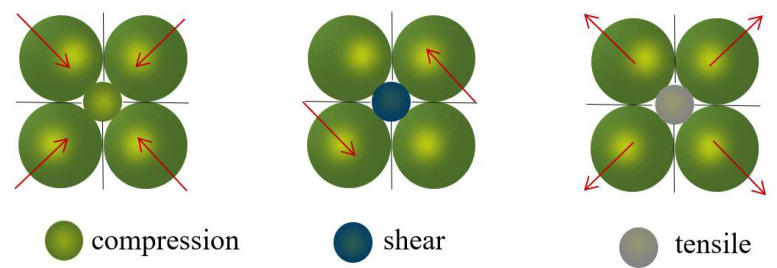
Schematic diagram of AE event failure mode in particle model.

**Figure 7 materials-15-06510-f007:**
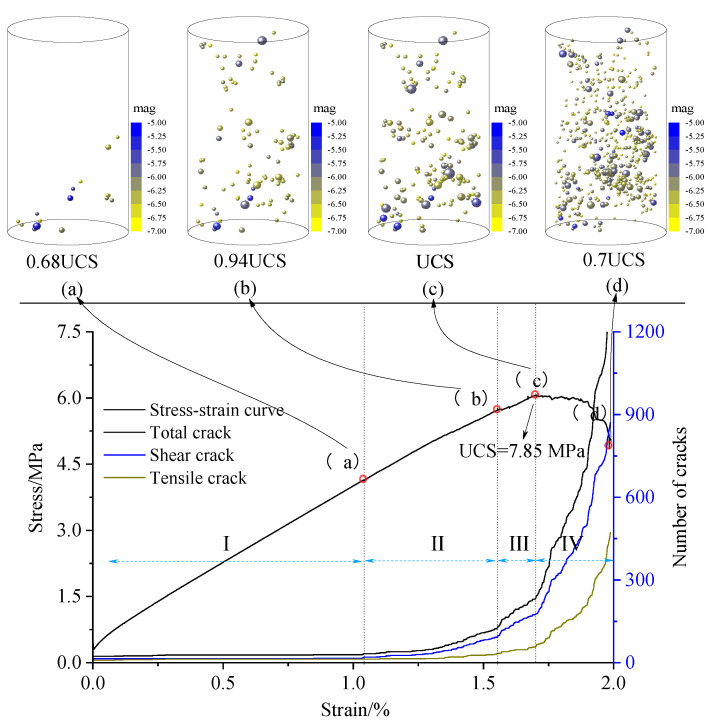
Stress–strain curve and number of cracks; upper snapshots show source distributions at different stages. I, II, III, and IV are the stages of crack generation, which are pore compaction stage, fracture expansion stage, peak stress stage and post peak stage, respectively.

**Figure 8 materials-15-06510-f008:**
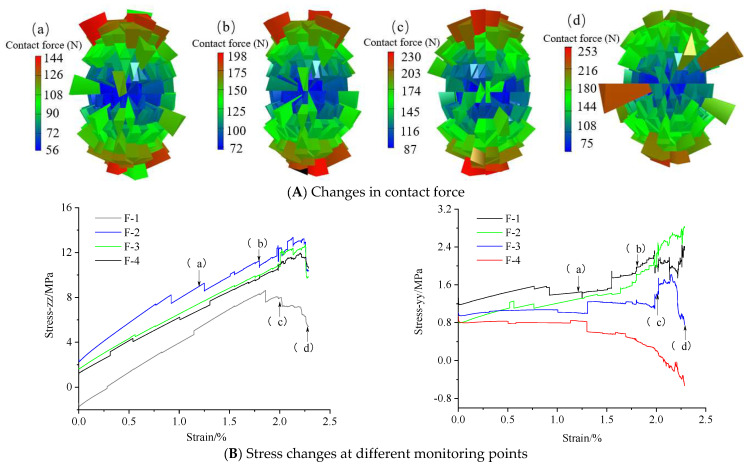
Changes in model internal stress and contact force. Panel (**A**) shows the internal contact force at different stages of the model. The warmer the color and the thicker the network, the greater the contact force. Panel (**B**) is the stress change at the monitoring point under the different stage corresponding to (**A**). (a–d) are the stages of crack generation, which are pore compaction stage, fracture expansion stage, peak stress stage and post peak stage, respectively.

**Figure 9 materials-15-06510-f009:**
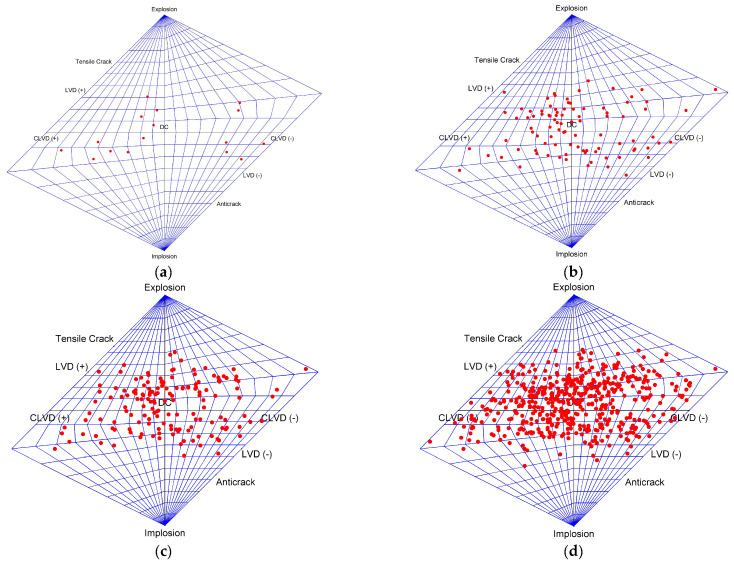
Hudson T–K locations of source models from (**a**–**d**).

**Figure 10 materials-15-06510-f010:**
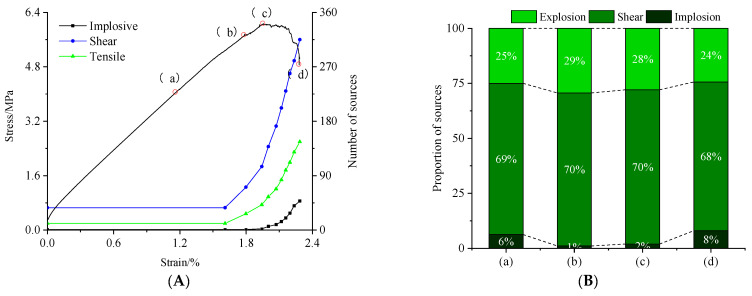
Nature of source versus time in uniaxial compression tests: (**A**) number and ratio of sources and (**B**) proportion of sources. (a–d) are the stages of crack generation, which are pore compaction stage, fracture expansion stage, peak stress stage and post peak stage, respectively.

**Figure 11 materials-15-06510-f011:**
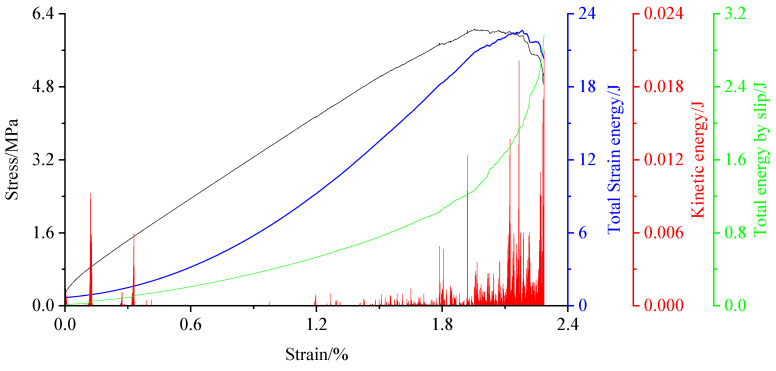
Various energy changes in coal failure process.

**Figure 12 materials-15-06510-f012:**
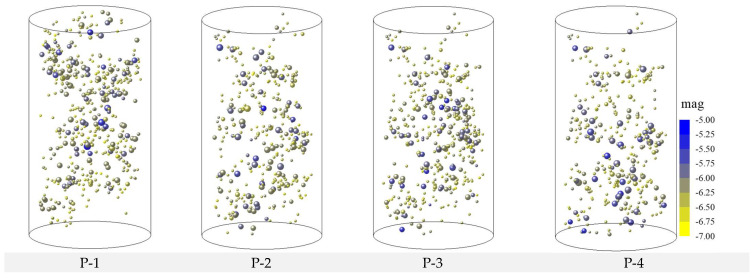
Source distributions of different models at different stages.

**Figure 13 materials-15-06510-f013:**
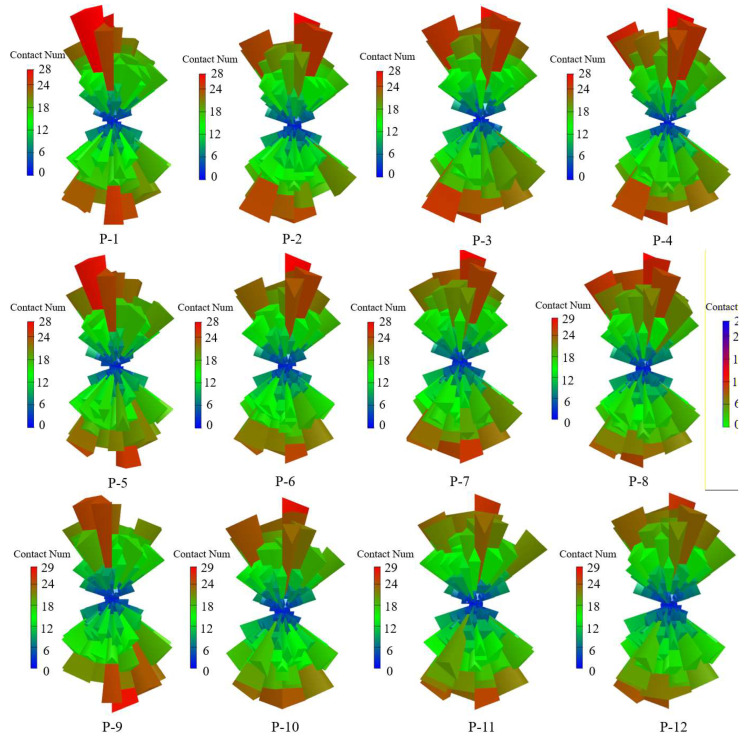
Number of contacts in different orientations.

**Figure 14 materials-15-06510-f014:**
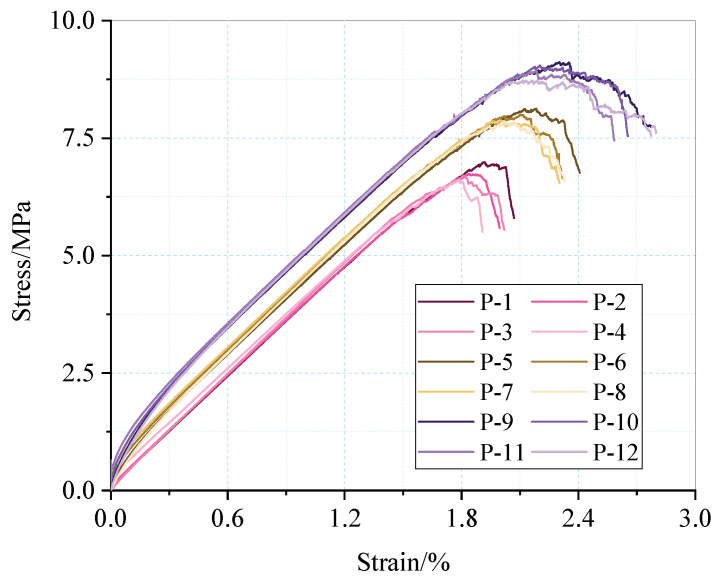
Stress–strain curves of different models.

**Figure 15 materials-15-06510-f015:**
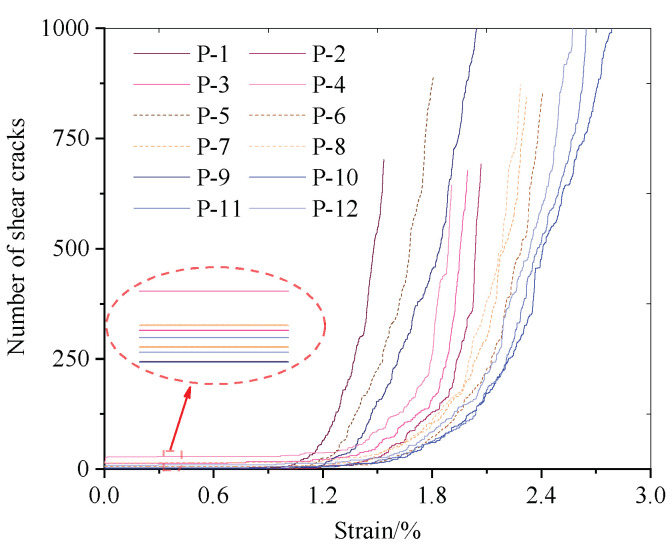
Development of cracks in coal body failure process in different models.

**Figure 16 materials-15-06510-f016:**
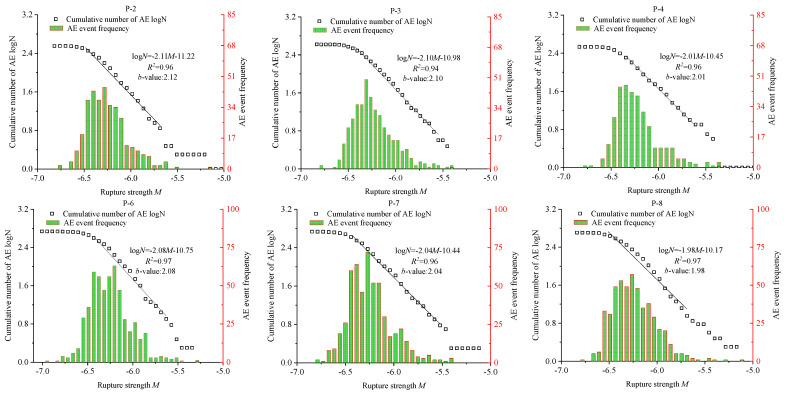
Frequency–amplitude curve and *b* value.

**Figure 17 materials-15-06510-f017:**
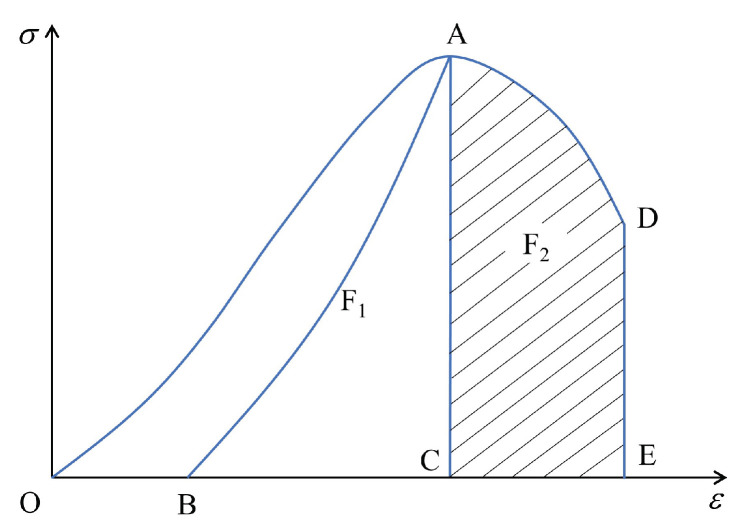
Relationship between dissipated and strain energies on stress–strain curve during uniaxial compression of coal. O, A and D correspond to the starting, peak stress and post peak value of the stress-strain curve respectively. B is the strain value of the model after unloading. C and D are the strain values corresponding to the peak and post peak stresses, respectively.

**Figure 18 materials-15-06510-f018:**
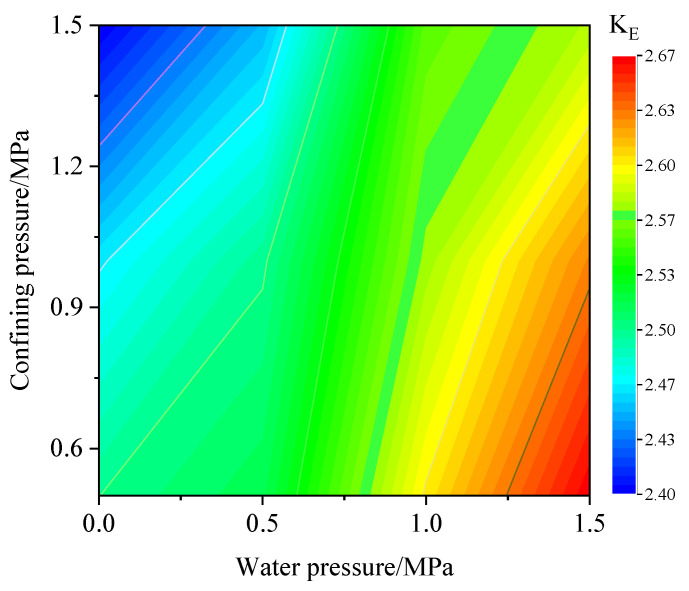
Change in *K_E_* under different water and confining pressures.

**Figure 19 materials-15-06510-f019:**
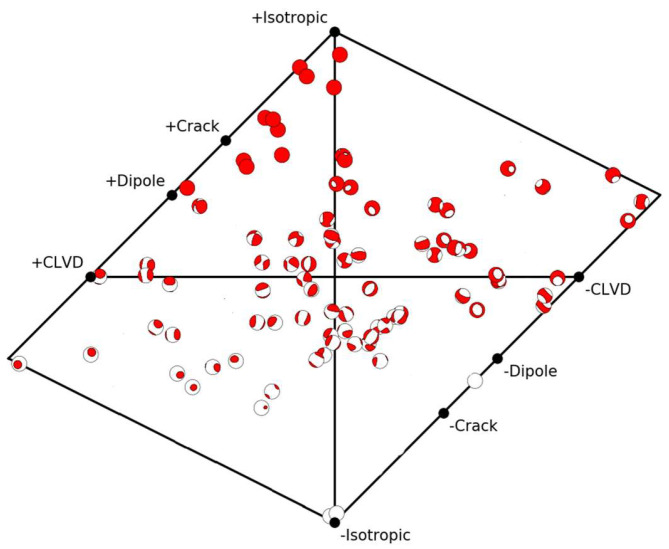
Hudson’s source type of coal failure under water pressure.

**Figure 20 materials-15-06510-f020:**
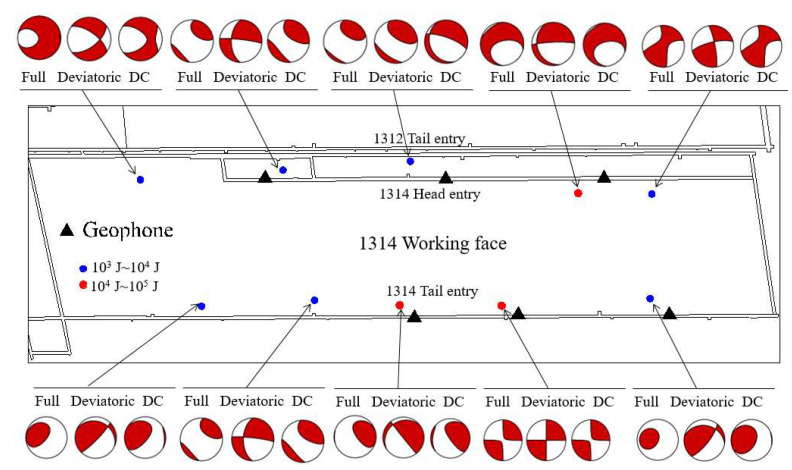
Beachballs in the 1314 working face.

**Figure 21 materials-15-06510-f021:**
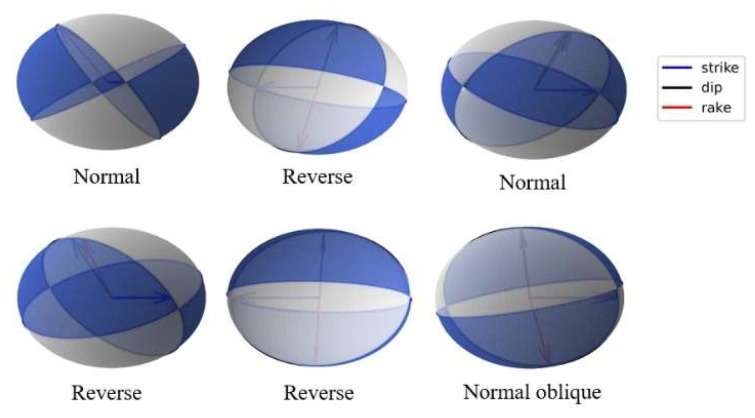
Fault plane solutions of the 1314 working face.

**Table 1 materials-15-06510-t001:** Microparameters of numerical model.

**Mechanical Parameters of Coal**			
Particle radius/mm	2.0–2.5	Elastic modulus/GPa	0.5
Density/kg·m^3^	1250	Friction angle/°	45
Cohesive strength/MPa	4.5	Porosity	0.18
**Parameters for Flow Model**			
Ap_zero/mm	1.3 × 10^−3^	Bulk_W/MPa	2.2 × 10^3^
Flow_perm/mm/s	7.0 × 10^−2^	Flow_dt/s	1.0 × 10^−3^
P_give/MPa	2.5	Gap_mul	0

**Table 2 materials-15-06510-t002:** Setting of confining and water pressures.

Model	P-1	P-2	P-3	P-4	P-5	P-6	P-7	P-8	P-9	P-10	P-11	P-12
Confining pressure/MPa	0.5	0.5	0.5	0.5	1	1	1	1	1.5	1.5	1.5	1.5
Water pressure /MPa	0	0.5	1.5	2.5	0	0.5	1.5	2.5	0	0.5	1.5	2.5

## Data Availability

The data in this manuscript are available from the authors.

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
