# Peer review of "Numerical and Field Investigations of Acoustic Emission Laws of Coal Fracture under Hydro-Mechanical Coupling Loading"

_materials, 2022, doi:10.3390/ma15196510_

Round 1

Reviewer 1 Report

The paper is well written and can be accepted for publication in the proposed form.

Detailed comments:

1.       Typos must be corrected;

2.       Enlarge some figures for better readability of the text on them;

3.       Improve the presentation and explanation of Figure 8;

4.       Study more recent references.

Author Response

Point 1: Typos must be corrected.

Response 1: Thanks for your good suggestions! We rechecked the language of the article again and corrected the typos mistakes. Meanwhile, to ensure the correctness of the language, so that readers better understand the research objectives and content, our manuscript has been edited by Elsevier. The Elsevier language editing service certificate is shown in the attachment.

Point 2: Enlarge some figures for better readability of the text on them.

Response 2: Thanks very much! According to your suggestion, I have readjusted the font size of some pictures to make them more clear to read.

Point 3: Improve the presentation and explanation of Figure 8.

Response 3: Thank you for your thoughtful suggestions! We have adjusted the legend in Fig.8 and described it in detail. The revised contents are as follows:

Fig. 8. Changes in model internal stress and contact force. (a) shows the internal contact force at different stages of the model. The warmer the color and the thicker the network means the greater the contact force. (b) is the stress change at the monitoring point under the different stage corresponding to (a).

By depicting the distribution of the micro contact force in the particle system, the macro mechanical mechanism can be revealed. Fig. 8 shows the contact force between particles in the model, which can better reflect the force in different directions of the model.

Point 4: Study more recent references.

Response 4: Thanks very much! We further supplemented the literature review content, explaining the differences and innovations from existing research. as follows:

Martyushev D A, Galkin S V, Shelepov V V. The Influence of the Rock Stress State on Matrix and Fracture Permeability under Conditions of Various Lithofacial Zones of the Tournaisian–Fammenian Oil Fields in the Upper Kama Region[J]. Moscow University Geology Bulletin, 2019, 74(6):573-581.

Riabokon E, Turbakov M, Popov N, et al. Study of the Influence of Nonlinear Dynamic Loads on Elastic Modulus of Carbonate Reservoir Rocks[J]. Energies, 2021, 14(24):8559-.

Reviewer 2 Report

In the presented article 1862940, the authors consider numerical and natural studies of acoustic-emission patterns of coal destruction during hydromechanical loading of couplings. The authors took coal under hydromechanical coupling as the object of study, the software for discrete elements PFC3D (particle flow code) was used to analyze the relationship between force, acoustic emission (AE) and energy during the destruction of coal. Based on the inversion of the moment tensor (MT), we revealed the distribution of AE events and the type of source during the initiation and propagation of a crack until the final destruction of coal. The authors made the following conclusions. (1) The coal body is mainly sheared during loading, which has been effectively confirmed by field surveys. (2) With an increase in water pressure and confining pressure, the number of both internal contacts and AE events increases, while the value of b AE decreases. (3) When the water pressure is constant, the kinetic energy (KE) decreases as the confining pressure increases. In contrast, when the confining pressure is held at a fixed level, KE increases as the water pressure increases. (4) Numerical simulation can effectively determine the microscopic mechanism of coal damage under various conditions.

Certainly, the work is interesting, scientific and has practical application. The authors would be encouraged to use a range of sources to help validate their findings.

The work may be published.

 Taking coal under hydro-mechanical coupling as the research object, the discrete element software, PFC3D, was used to analyse the relationships among the force, acoustic emission, and energy during coal fracture. The topic is relevant to this field and complements previous research with better measurements and analysis of results. The conclusions are consistent with the arguments and results presented in the paper and give a good presentation of the topic. References are appropriate, but they can be supplemented with newer ones.

Martyushev, D.A., Galkin, S.V., Shelepov, V.V. The influence of the rock stress state on matrix and fracture permeability under conditions of various lithofacial zones of the tournaisian-fammenian oil fields in the Upper Kama Region. Moscow University Geology Bulletin. 2019. V.74(6). pp.573-581. https://doi.org/10.3103/S0145875219060061

Riabokon, Evgeniia, Turbakov, Mikhail, Popov, Nikita, Kozhevnikov, Evgenii, Poplygin, Vladimira, Guzev, Mikhail Study of the influence of nonlinear dynamic loads on elastic modulus of carbonate reservoir rocks. Energies, 2021, 14(24), 8559. https://doi.org/10.3390/en14248559

Author Response

Response to Reviewer 2 Comments

Point 1:  Taking coal under hydro-mechanical coupling as the research object, the discrete element software, PFC3D, was used to analyse the relationships among the force, acoustic emission, and energy during coal fracture. The topic is relevant to this field and complements previous research with better measurements and analysis of results. The conclusions are consistent with the arguments and results presented in the paper and give a good presentation of the topic. References are appropriate, but they can be supplemented with newer ones.

Response 1: Thank you for your thoughtful suggestions! We further supplemented the literature review content, explaining the differences and innovations from existing research. Meanwhile, in order to make the introduction more complete, according to your suggestion, Some contents were added.

In the introduction, the research status and the basis for the research are summarized and added:

Many scholars have studied the changes of rock mechanical parameters under the hydro-mechanical coupling loading in the laboratory, but there are few studies on the influence of seepage under different water and confining pressure. In addition, few scholars focused on the source rupture type in the process of rock seepage through AE.

Martyushev et al. (2019) studied the rock stress state and fracture permeability under conditions of various lithofacial zones. Most scholars widely use AE to locate the crack position of rock, and then study the force of the specimen (Riabokon et al. 2021). However, there are few researches on the focal mechanism of AE, and the analysis on the rupture form of rock/coal mass under different stress conditions is still lacking.

Besides, it is consistent with the results of our field investigation. These results are highly significant for guiding the safe production of coal mines.

The added literature:

[1] Martyushev D A, Galkin S V, Shelepov V V. The Influence of the Rock Stress State on Matrix and Fracture Permeability under Conditions of Various Lithofacial Zones of the Tournaisian–Fammenian Oil Fields in the Upper Kama Region[J]. Moscow University Geology Bulletin, 2019, 74(6):573-581.

[2] Riabokon E, Turbakov M, Popov N, et al. Study of the Influence of Nonlinear Dynamic Loads on Elastic Modulus of Carbonate Reservoir Rocks[J]. Energies, 2021, 14(24):8559-.

Reviewer 3 Report

This paper investigates acoustic emission laws of coal fracture under hydro-me-chanical coupling loading. The idea is interesting and present very interesting results for such a topic. Authors need to give more details about the employed theoretical and possible numerical approaches so that anyone can reproduce the results easily. More details about validating the current investigations and possible comparisons with other results in literature is also needed. The introduction also requires more information about the current work and how this is connected to other researchers in the field. Authors should improve the ABSTRACT AND CONCLUSION sections i.e. to describe the novelty of the work, what is the current simulation future directions and how this can be further validated. Specifically, conclusions are drawn in the abstract which shouldn’t be the case.

Author Response

Response to Reviewer 3 Comments

Point 1: This paper investigates acoustic emission laws of coal fracture under hydro-me-chanical coupling loading. The idea is interesting and present very interesting results for such a topic. Authors need to give more details about the employed theoretical and possible numerical approaches so that anyone can reproduce the results easily. More details about validating the current investigations and possible comparisons with other results in literature is also needed. The introduction also requires more information about the current work and how this is connected to other researchers in the field. Authors should improve the ABSTRACT AND CONCLUSION sections i.e. to describe the novelty of the work, what is the current simulation future directions and how this can be further validated. Specifically, conclusions are drawn in the abstract which shouldn’t be the case.

Response 1: Thank you for your thoughtful suggestions! We have revised the abstract and introduction. In the abstract, the current work and research novelty were summarized. In the introduction, the current research status and the basis of the next research are stated. In Chapter 3.2, we added the description of the simulation process so that anyone can reproduce the results easily. The revised contents are as follows:

Abstract:Taking coal under hydro-mechanical coupling as the research object, the discrete element software, PFC3D (particle flow code), was used to analyse the relationships among the force, acoustic emission (AE), and energy during coal rupture. Based on the moment tensor (MT) inversion, we revealed the AE event distribution and source type during crack initiation and propagation until the final failure of coal. Meanwhile, we examined the relationships among the stress, number and type of cracks, magnitude, KE, and b value of AE under different water and confining pressures. The results show that the numerical simulation can effectively determine the microscopic damage mechanism of coal under different conditions. Moreover, the rupture type of the numerical simulation is consistent with the field investigations, which verifies the rationality of the simulation. These research results can provide reference for safety production evaluation of water inrush mines.

In the introduction, the research status and the basis for the research are summarized and added:

Many scholars have studied the changes of rock mechanical parameters under the hydro-mechanical coupling loading in the laboratory, but there are few studies on the influence of seepage under different water and confining pressure. In addition, few scholars focused on the source rupture type in the process of rock seepage through AE.

Martyushev et al. (2019) studied the rock stress state and fracture permeability under conditions of various lithofacial zones. Most scholars widely use AE to locate the crack position of rock, and then study the force of the specimen (Riabokon et al. 2021). However, there are few researches on the focal mechanism of AE, and the analysis on the rupture form of rock/coal mass under different stress conditions is still lacking.

Besides, it is consistent with the results of our field investigation. These results are highly significant for guiding the safe production of coal mines.

Chapter 3.2:

To study the mechanical characteristics and instability mechanism of coal under the action of seepage, a model was set-up, which is shown in Fig. 2. For the model the standard size (Φ50 mm × 100 mm) was adopted, and a total of 3117 particles were generated. The radius was 2–2.5 mm. The linear contact bond model was used between particles, and the servo mechanism was used to apply different confining pressures around the model. When the confining pressure reached the target stress, the particles in contact with the wall were fixed. A high-pressure water area was applied within the range of 0–10 mm in the z-axis direction at the bottom of the model. Under the action of the pressure difference, a water body seeped from the bottom to the top of the model. At this time, the water pressure is close to linear distribution along the longitudinal direction, which can be regarded as the equilibrium state of hydro-mechanical coupling. After the balance was reached, the upper and lower walls were controlled to apply an axial pressure at a speed of 0.03 m/s, and the loading was stopped when the residual stress reached 70% peak stress.

The model parameter calibration was based on the uniaxial compressive strength, strain, and failure form of the specimen. Based on Fig. 3, the peak stresses from the experiments and numerical simulations are 6.09 MPa and 6.05 MPa, respectively and the corresponding strains are 1.93% and 1.95%. The difference between them is relatively small, and the failure modes are also similar. From the variation trends of the AE events, the microscopic properties inside the coal body are changed owing to the softening effect of water. When the strain is approximately 5%, the number of AE events shows remarkable fluctuations, which is followed by a quiet period. When the strain reaches 1.6%, the AE events abruptly increase, and subsequently cracks gradually penetrate to form macroscopic cracks, and finally, the number of AE reaches the maximum. Based on above analysis, the rationality of the micro-parameter values can be verified, which are listed in Table 1.

Added literature:

[1] Martyushev D A, Galkin S V, Shelepov V V. The Influence of the Rock Stress State on Matrix and Fracture Permeability under Conditions of Various Lithofacial Zones of the Tournaisian–Fammenian Oil Fields in the Upper Kama Region[J]. Moscow University Geology Bulletin, 2019, 74(6):573-581.

[2] Riabokon E, Turbakov M, Popov N, et al. Study of the Influence of Nonlinear Dynamic Loads on Elastic Modulus of Carbonate Reservoir Rocks[J]. Energies, 2021, 14(24):8559-.

Round 2

Reviewer 3 Report

Authors have modified their interesting paper as suggested. Accept it as it is.